# Brain Temperature Measured by Magnetic Resonance Spectroscopy to Predict Clinical Outcome in Patients with Infarction

**DOI:** 10.3390/s21020490

**Published:** 2021-01-12

**Authors:** Tomohisa Ishida, Takashi Inoue, Tomoo Inoue, Toshiki Endo, Miki Fujimura, Kuniyasu Niizuma, Hidenori Endo, Teiji Tominaga

**Affiliations:** 1Department of Neurosurgery, National Hospital Organization Sendai Medical Center, 2-11-12 Miyagino, Sendai, Miyagi 983-8520, Japan; tomy.ixxx4@gmail.com (T.I.); tomoo49@gmail.com (T.I.); 2Department of Neurosurgery, Kohnan Hospital, 4-20-1 Nagamachiminami, Sendai, Miyagi 982-0012, Japan; endo@nsg.med.tohoku.ac.jp (T.E.); fujimur@nsg.med.tohoku.ac.jp (M.F.); 3Department of Neurosurgery, Tohoku University Graduate School of Medicine, 2-1 Seiryomachi, Sendai, Miyagi 980-8575, Japan; niizuma@nsg.med.tohoku.ac.jp (K.N.); hideendo@gmail.com (H.E.); tomi@nsg.med.tohoku.ac.jp (T.T.)

**Keywords:** acute ischemic stroke, brain temperature, magnetic resonance spectroscopy, less invasive, clinical outcomes, cerebral blood flow change

## Abstract

Acute ischemic stroke is characterized by dynamic changes in metabolism and hemodynamics, which can affect brain temperature. We used proton magnetic resonance (MR) spectroscopy under everyday clinical settings to measure brain temperature in seven patients with internal carotid artery occlusion to explore the relationship between lesion temperature and clinical course. Regions of interest were selected in the infarct area and the corresponding contralateral region. Single-voxel MR spectroscopy was performed using the following parameters: 2000-ms repetition time, 144-ms echo time, and 128 excitations. Brain temperature was calculated from the chemical shift between water and *N*-acetyl aspartate, choline-containing compounds, or creatine phosphate. Within 48 h of onset, compared with the contralateral region temperature, brain temperature in the ischemic lesion was lower in five patients and higher in two patients. Severe brain swelling occurred subsequently in three of the five patients with lower lesion temperatures, but in neither of the two patients with higher lesion temperatures. The use of proton MR spectroscopy to measure brain temperature in patients with internal carotid artery occlusion may predict brain swelling and subsequent motor deficits, allowing for more effective early surgical intervention. Moreover, our methodology allows for MR spectroscopy to be used in everyday clinical settings.

## 1. Introduction

Brain thermoregulation depends on a delicate balance between heat-producing and heat-dissipating mechanisms [1,2,3]. Fundamentally, brain temperature in the healthy human at rest is determined by the balance between heat produced by cerebral energy turnover, which is identical to cerebral metabolism, and heat that is removed, primarily by cerebral blood flow [4,5,6]. The disturbance of this homeostasis following central nervous system injury, such as cerebrovascular ischemia, may cause progressive ischemic damage to neuronal substrates because of higher brain temperatures [7,8]; this may lead to more severe motor deficits, among other impairments. For example, higher brain temperatures in the affected regions of patients with unilateral internal carotid stenosis have been reported to predict hyperperfusion after carotid endarterectomy, with related motor weakness [9]. Therefore, monitoring and therapeutic modulation of brain temperature after acute ischemic stroke are important for assessing clinical conditions and treatment options, to avoid severe motor dysfunction and death.

Brain swelling associated with large ischemic areas can cause severe motor deficits and may be life threatening [10]. Early predictors of brain swelling are urgently needed because decompressive surgery before the occurrence of clinical deterioration can lead to better outcomes [11,12]; however, to date, no such early predictors have been established [6]. Although early elevations in brain temperature are known to indicate poor neurological outcomes [13,14], the relationship between brain temperature and brain swelling remains unclear and requires further investigation.

In human patients, noninvasive methods for monitoring brain temperature in clinical situations are needed [15]. Proton magnetic resonance (MR) spectroscopy has been demonstrated to accurately measure brain temperature in both animals [16,17,18] and humans [9,13,14,16,19,20,21]. In this technique, chemical shifts of water protons are measured against an abundant internal reference, such as *N*-acetyl aspartate (NAA) [22], choline, and/or creatine [23,24,25], allowing for relatively accurate measurements of brain temperature [15]. However, for various reasons, MR spectroscopy is not commonly used to measure brain temperature in everyday clinical practice. For example, although a recent study reported that calibration measurements are important for accurate brain temperature estimation when using this approach [26], their application is impractical in single patients. Thus, to assess the clinical value of this technique, methods that can enable its use under everyday conditions are needed.

In the present study, we aimed to investigate the use of proton MR spectroscopy to noninvasively measure brain temperatures in patients with severe ischemic stroke using a relatively simple method that is appropriate for everyday clinical use. Our preliminary findings suggest that this technology might be useful for predicting clinical courses, including motor function, in such patients. In the future, this technique could be used to indicate severe brain swelling before it occurs, thus allowing for earlier treatment and better patient prognosis.

## 2. Subjects and Methods

### 2.1. Patient Characteristics

This study included patients with internal carotid artery occlusion who were admitted to our hospital within 24 h from the onset and could have undergone MR spectroscopy within 48 h. One female and six male patients (aged 57–88 years) with internal carotid artery occlusion in the acute stage and National Institutes of Health Stroke Scale scores of 18 to 37 (median 25.7) were enrolled. The modified Rankin Scale was used to measure the degree of disability in each patient at discharge, or within 5 days after admission in the patients who died before discharge. MR imaging was obtained at 15 to 48 h (median 24 h) after onset. The time of onset was taken as the first occurrence of symptoms; if the patient awoke with stroke, then the time of onset was taken as the time that the patient was last known to be symptom-free. The study protocol was approved by the Ethics Committee of Kohnan Hospital (approval number Rin2010-01) and performed in accordance with the guidelines of the latest version of the Declaration of Helsinki. Written informed consent was given by each patient’s family prior to the study.

### 2.2. MR Imaging

All MR imaging was performed using a Signa VH/i 3.0 T MR imaging system (General Electric Medical Systems, Milwaukee, WI, USA) and parallel imaging head coil. Imaging with a 3.0 T system has been previously reported to give good accuracy [27,28]. For T2-weighted MR imaging, the short inversion time inversion recovery sequence was used with the following parameters: repetition time (TR), 4000 ms; echo time (TE), 81 ms; inversion time, 100 ms; matrix, 512 × 384; field of view, 400 mm; and 3.5-mm slice thickness. Single-voxel MR spectroscopy was performed with the following parameters: TR 2000 ms, TE 144 ms, and 128 excitations. A point-resolved spectroscopy pulse sequence was used. The voxels of interest, with a size of 20 mm × 20 mm × 30 mm, were set in the infarct area and the contralateral area of each patient, as identified on T2-weighted images (Figure 1). The image acquisition took 15 min (not including patient settling time), and the environmental temperature was maintained at 21 °C to 25 °C. The MR spectra included peaks assigned to choline-containing compounds at 3.2 parts per million (ppm), creatine phosphate at 3.0 ppm, and NAA at 2.0 ppm. All spectra were inspected visually and discarded if judged to be of poor quality (i.e., if the signal-to-noise ratio was such that peaks were unable to be detected). Brain temperature was calculated from the chemical shift between water and choline-containing compounds, creatine phosphate, or NAA; the most recognizable of the three peaks was used for each measurement. To calculate the brain temperature, a 0.01 ppm shift in the water peak was taken as a 1 °C change [6,9,29]. The ratio of brain temperature in the affected hemisphere to that in the contralateral hemisphere was then calculated for each patient, as reported previously [6,9].

## 3. Results

The mean brain temperature, as measured by MR spectroscopy, was 38.7 °C ± 13.7 °C (mean ± standard deviation). Mean systemic temperature measured by an electronic clinical thermometer in the armpit was 37.4 °C ± 0.5 °C. Table 1 shows the clinical summaries and brain temperature ratios of the seven patients. Two patients had a higher brain temperature on the pathological side than on the contralateral side (i.e., a ratio above 1), and neither of these patients had subsequent severe brain swelling. The other five patients had a lower brain temperature on the pathological side than on the contralateral side (i.e., a ratio below 1). Three of these five patients died of severe brain swelling with excessive midline shift. The *p* value was 0.1473 in the Chi-square test. In all patients, brain temperatures were measured before any severe brain swelling or clinical signs of herniation were detected.

## 4. Discussion

The present study demonstrated that brain temperature can be measured noninvasively in patients in the acute stage of severe ischemic stroke using MR thermometry under everyday clinical conditions. Some patients had higher brain temperatures in the stroke lesion compared with the contralateral region, while other patients had lower brain temperatures in the stroke lesion. Notably, all patients who subsequently suffered herniation had lower brain temperatures in the stroke lesion compared with the contralateral region.

Brain temperature may theoretically be elevated in hypoperfused but viable tissues, such as the ischemic penumbra, if the ischemic tissue is metabolically active but the clearance of heat is insufficient [7,30,31]. Brain temperature gradually decreases with the development of more mature areas of infarction in which hypoperfusion has resulted in metabolically inert tissues [14,19]. In a previous study, brain temperatures were measured both invasively, using a sensor passed through a burr hole or craniotomy, and noninvasively, using MR spectroscopy. Brain temperature in the infarcted hemisphere was higher than that in the contralateral hemisphere at 6 h after middle cerebral artery infarction, but gradually became lower than that in the contralateral hemisphere after 12 h [16]. In the present study, a noninvasive measurement of the ischemic core lesion also demonstrated predominantly lower temperatures in patients between 15 and 48 h after infarction.

Interestingly, in the present study, all three patients who subsequently underwent herniation caused by severe brain swelling had a lower brain temperature in the lesion compared with the contralateral region. The lower brain temperature in the lesion was observed before the occurrence of severe brain swelling or any clinical signs of herniation. Similarly, direct monitoring has previously revealed that brain temperatures fall below bladder temperatures some hours before the occurrence of bilateral pupil dilation and fixation, or clinical signs of herniation [19]. This finding may be supported by other biological evidence. For example, brain swelling is caused by both angioedema and cellular edema [10,11,12]. Such edema might occur with local reperfusion from the collateral vessels, and may cause a decline in brain temperature as a result of the heat clearance effect. Thus, decreases in temperature may occur earlier in the presence of brain edema. The observation of temperature changes in the acute phase of ischemia may therefore be an early indicator of the risk of brain herniation and may lead to more effective surgical intervention. The present finding of a possible relationship between brain temperature and brain swelling after ischemic stroke may, therefore, have important implications for its treatment and might lead to better motor outcomes in patients.

Although many clinical studies have used MR spectroscopy to measure brain temperatures under strict experimental protocols [9,13,14,16,19,20], the present study focused on the use of this technique in everyday clinical practice, where many factors are unable to be controlled. Specifically, we used a relatively common 3.0 T MR imaging system, with a short imaging time, small voxel size, and simple calculation method for clinical usage. We were also flexible with regard to the time period between infarction and temperature measurement and used a temperature ratio (rather than the more commonly used temperature difference) as a simple control for external factors. Taken together, these techniques will facilitate the use of MR spectroscopy for measuring brain temperature in everyday clinical practice.

Several limitations of this study should be acknowledged. The small sample size is a major limitation, and a larger study is needed to confirm our preliminary findings. Sequential measurements should be made in each patient to investigate changes in brain temperature over time. In addition, in the present study, although the average brain temperature was similar to that reported via direct measurements of neurosurgical patients [32], the measured brain temperatures varied widely, and accurate temperature readings were difficult to confirm, similar to reports in traumatically injured brain tissue [8]. This may be because MR thermometry can be affected by the ambient temperature and other factors, such as pH. Moreover, MR spectra may change during imaging because of dynamic changes in both metabolism and hemodynamics in the acute phase [6]. In an attempt to control for these factors, we calculated the ratio of the pathological side of the brain to the contralateral side of the brain to reduce variance. This simple method of controlling for some outside factors means that our technique is better suited to clinical use, where such factors are often unable to be controlled directly.

In conclusion, changes in brain temperature in stroke lesions can be measured using MR spectroscopy under everyday clinical conditions. A lower brain temperature in the lesion compared with the contralateral region may indicate a higher risk of subsequent brain swelling in patients with internal carotid artery occlusion, although further studies are required. Accurately predicting future brain swelling is important because it will allow for more effective early surgical intervention, thereby potentially lowering both mortality and the severity of motor deficits in patients.

## Figures and Tables

**Figure 1 sensors-21-00490-f001:**
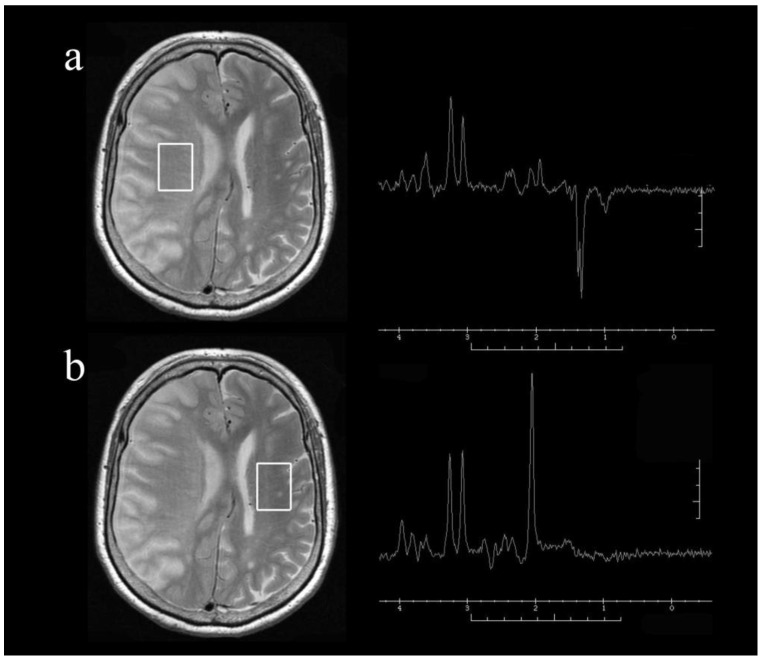
Representative axial T2-weighted magnetic resonance images and single-voxel proton magnetic resonance spectra obtained from the voxels of interest (square areas) in the infarct hemisphere (**a**) and normal hemisphere (**b**) of a patient with right internal carotid artery occlusion at 24 h after onset. Note the presence of lactate with relatively preserved *N*-acetyl aspartate (NAA) in the infarct hemisphere.

**Table 1 sensors-21-00490-t001:** Summary of clinical findings in seven patients with internal carotid artery occlusion.

Case No.	Age (Years)	Sex	Side of Occlusion	Brain Temperature Ratio ^1^	Time from Onset (h)	Severe Brain Swelling (−/+)	mRS	Systemic Temp. (°C)	NIHSS	Time to Death/Discharge (Days)
1	72	Male	Rt	1.5	31	−	5	37.8	18	50
2	57	Male	Rt	1.23	28	−	5	37.5	29	27
3	88	Male	Lt	0.91	24	+	6	37.2	37	5
4	69	Male	Lt	0.8	48	−	5	37.6	27	54
5	87	Female	Rt	0.78	24	+	6	37.3	21	4
6	84	Male	Rt	0.71	21	+	6	37.4	30	3
7	70	Male	Rt	0.67	15	−	5	37.1	18	62

^1^ Brain temperature ratio: brain temperature in the ischemic lesion/contralateral region. Abbreviations: Lt, left; mRS, modified Rankin scale; NIHSS, National Institutes of Health Stroke Scale; Rt, right; temp., temperature.

## Data Availability

The data presented in this study are available on request from the corresponding author. The data are not publicly available for reasons of privacy.

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
