# Peer review of "Brain Temperature Measured by Magnetic Resonance Spectroscopy to Predict Clinical Outcome in Patients with Infarction"

_sensors, 2021, doi:10.3390/s21020490_

Round 1

Reviewer 1 Report

The objectives of the study were to measure brain temperature of patients with internal carotid artery occlusion and to explore the relationship between lesion temperature and clinical course using proton magnetic resonance (MR) spectroscopy in daily clinical setting. The study found that brain temperature in ischemic lesion was lower in 5 and higher in 2 out of 7 patients and that severe brain swelling existed in the 3 out of 5 with ischemic lesion but not of the 2 patients with higher lesion temperature. And thus, the study proposed that changes in brain temperature in stroke lesions can be observed and measured using MR spectroscopy, and this daily clinical approach may predict brain swelling and subsequent motor deficits. 

The work has been well designed and written. There are a few comments to be addressed before publications.

1. Line 84: It was mentioned that "Written informed 84
consent was given by each patient’s family prior to the study." However, Line 210: "Informed Consent Statement: Informed consent was obtained from all subjects involved in the study." Please clarify.

2. Please include the inclusion and exclusion criteria of the patients selection and describe comorbidities, if any, in the 2.1 section.

3. Line 170: "the present study focused
on the use of this technique in everyday clinical practice, where many factors are unable 
to be controlled." Please provide examples of what factors may not be able to be controlled and how it could have been improved for future study.

Author Response

  • The first comment was about “Written informed consent” of this study. We took the written informed consent from the patients’ family because of the patients had sever conscious disturbance. We mistook in Line 210 (previous version), so we revised this sentence. (Line 215)
  • The second comment was that we should include the inclusion and exclusion criteria of this study. We agree with this comment so we added the sentences in 2.1 section. We selected the patients with internal carotid artery occlusion who were taken MR spectroscopy in acute stage. (Line 75-79)
  • The third comment was what is the “uncontrolled factors” in our limitation. We focused the clinical usage of MR spectroscopy, so we couldn’t have enough scan time for patients. We also scanned MR spectroscopy at various time from the onset. We added the sentence in the limitation for more readers’ understanding. We hope these limitations should be taken over by large subjects’ number future studies. (Line 176, 182)

Reviewer 2 Report

This is a very important manuscript and here are a few suggestions to help improve the manuscript.

1) perform a cross correlation between brain temperature and clinical severity scale or regression analysis

2)create a scatter plot of the results of the correlation complete with regression line and p value

3) whenever the authors state that there is relationship between brain temperature and clinical outcome then statistics should be presented.

Author Response

  • The first comment was that we should perform cross correlation or regression analysis between brain temperature and clinical severity. We agree with this comment but unfortunately, we had only seven subjects in this study. We performed the cross-correlation analysis: the r-square was 0.1605, p=0.2155. We didn’t describe this value in our manuscript.
  • The second comment was that we should create a scatter plot and show p value of the results. This idea is very important for the understandings of our study if we would have taken larger number of subjects. The p value was 0.2155.
  • The third comment was that we should present statistics in our manuscript. This comment is also reliable for the scientific papers. We performed the chi-square test between brain swelling and brain temperature elevation against contralateral side. The p value was 0.1473, and we consider that this result depends on small sample size. We added this p value in Results section. (Line 130)